# Mutation in *ATG5* reduces autophagy and leads to ataxia with developmental delay

Myungjin Kim[1†], Erin Sandford[2†], Damian Gatica[3,4], Yu Qiu[5], Xu Liu[3,4], Yumei Zheng[5], Brenda A Schulman[5,6], Jishu Xu[7], Ian Semple[1], Seung-Hyun Ro[1], Boyoung Kim[1], R Nehir Mavioglu[8], Aslıhan Tolun[8], Andras Jipa[9,10], Szabolcs Takats[10], Manuela Karpati[10], Jun Z Li[7,11], Zuhal Yapici[12], Gabor Juhasz[9,10], Jun Hee Lee[1*], Daniel J Klionsky[3,4*], Margit Burmeister[2,7,11,13*]

[1]Department of Molecular and Integrative Physiology, University of Michigan, Ann Arbor, United States; [2]Molecular and Behavioral Neuroscience Institute, University of Michigan, Ann Arbor, United States; [3]Department of Molecular, Cellular, and Developmental Biology, University of Michigan, Ann Arbor, United States; [4]Life Sciences Institute, University of Michigan, Ann Arbor, United States; [5]Department of Structural Biology, St Jude Children's Research Hospital, Memphis, United States; [6]Howard Hughes Medical Institute, St. Jude Children's Research Hospital, Memphis, United States; [7]Department of Human Genetics, University of Michigan, Ann Arbor, United States; [8]Department of Molecular Biology and Genetics, Boğaziçi University, Istanbul, Turkey; [9]Institute of Genetics, Biological Research Centre, Hungarian Academy of Sciences, Szeged, Hungary; [10]Department of Anatomy, Cell and Developmental Biology, Eötvös Loránd University, Budapest, Hungary; [11]Department of Computational Medicine and Bioinformatics, University of Michigan, Ann Arbor, United States; [12]Department of Neurology, Faculty of Medicine, Istanbul University, Istanbul, Turkey; [13]Department of Psychiatry, University of Michigan, Ann Arbor, United States

*For correspondence: leeju@ umich.edu (JHL); klionsky@umich. edu (DJK); margit@umich.edu (MB)

[†]These authors contributed equally to this work

Competing interests: The authors declare that no competing interests exist.

**Abstract** Autophagy is required for the homeostasis of cellular material and is proposed to be involved in many aspects of health. Defects in the autophagy pathway have been observed in neurodegenerative disorders; however, no genetically-inherited pathogenic mutations in any of the core autophagy-related (*ATG*) genes have been reported in human patients to date. We identified a homozygous missense mutation, changing a conserved amino acid, in *ATG5* in two siblings with congenital ataxia, mental retardation, and developmental delay. The subjects' cells display a decrease in autophagy flux and defects in conjugation of ATG12 to ATG5. The homologous mutation in yeast demonstrates a 30-50% reduction of induced autophagy. Flies in which Atg5 is substituted with the mutant human ATG5 exhibit severe movement disorder, in contrast to flies expressing the wild-type human protein. Our results demonstrate the critical role of autophagy in preventing neurological diseases and maintaining neuronal health.

## Introduction

Macroautophagy, referred to hereafter as autophagy, is a cellular process by which proteins and organelles are degraded and recycled through sequestration within autophagosomes and delivery to lysosomes (*Levine and Klionsky, 2004*). The autophagy pathway is highly conserved and required for organismal development and function. Defects in autophagy are associated with diseases including cancer, metabolic disruption, and neurodegenerative disorder (*Choi et al., 2013*;

**eLife digest** Ataxia is a rare disease that affects balance and co-ordination, leading to difficulties in walking and other movements. The disease mostly affects adults, but some children are born with it and they often have additional cognitive and developmental problems. Mutations in at least 60 genes are known to be able to cause ataxia, but it is thought that there are still more to be found.

Kim, Sandford et al. studied two siblings with the childhood form of ataxia and found that they both had a mutation in a gene called *ATG5*. The protein produced by the mutant *ATG5* gene was less able to interact with another protein called ATG12. Furthermore, the cells of both children had defects in a process called autophagy – which destroys old and faulty proteins to prevent them accumulating and causing damage to the cell.

Next, Kim, Sandford et al. examined the effect of this mutation in baker's yeast cells. Cells with a mutation in the yeast equivalent of human *ATG5* had lower levels of autophagy than normal cells. Further experiments used fruit flies that lacked fly *Atg5*, which were unable to fly or walk properly. Inserting the normal form of human *ATG5* into the flies restored normal movement, but the mutant form of the gene had less of an effect.

These findings suggest that a mutation in *ATG5* can be responsible for the symptoms of childhood ataxia. Kim, Sandford et al. think that other people with severe ataxia may have mutations in genes involved in autophagy. Therefore, the next step is to study autophagy in cells from many other ataxia patients.

*Cuervo and Wong, 2014*; *Frake et al., 2015*). Patients with mutations in any of the non-redundant core autophagy-related (*ATG*) genes have not previously been reported.

Ataxia is a neurodegenerative disease caused by disruption of the cerebellum and Purkinje cells, which results in the lack of coordinated muscle movements. Large phenotype diversity is present in individuals with ataxia, including age of onset, rate of progression, and other accompanying neurological and non-neurological features (*Jayadev and Bird, 2013*), with corresponding genotypic heterogeneity (*Sandford and Burmeister, 2014*). Even within the more defined phenotype of childhood ataxia with developmental delay, there are a large number of associated genes, such that similar phenotypic features alone are often insufficient information for an accurate diagnosis (*Burns et al., 2014*; *De Michele and Filla, 2012*; *Jayadev and Bird, 2013*). Identification of genetic causes of childhood ataxia is important for understanding disease pathogenesis and for possible future treatment development.

Whole exome sequencing has been successfully utilized to identify known and novel genetic mutations responsible for ataxia (*Burns et al., 2014*; *Fogel et al., 2014*). Identification of candidate genes can be further verified through additional molecular analysis and utilization of specific and general animal models. Here we identified a novel mutation in a core autophagy gene, *ATG5*, in two children with ataxia, and demonstrate a reduction in autophagic response, also reproducing the phenotype in yeast and fly models.

## Results

### E122D mutation in ATG5 is associated with familial ataxia

Two Turkish siblings presented with ataxia and developmental delay in childhood, as previously described (*Yapici and Eraksoy, 2005*). We performed linkage analysis on both affected siblings, their unaffected siblings, and their unaffected mother, using a model of remote parental consanguinity and identified a single broad (>14 Mbp) peak with LOD score 3.16 on chromosome 6q21, between 102 and 116 Mb (*Figure 1*). Whole exome sequencing identified a homozygous missense mutation, hg19 chr6:106,727,648 T>A, corresponding to E122D in *ATG5* (*Figure 2A*) as the only damaging mutation within the genetically identified chromosomal linkage interval. The mutation was Sanger verified and found absent from variant databases and from Turkish controls.

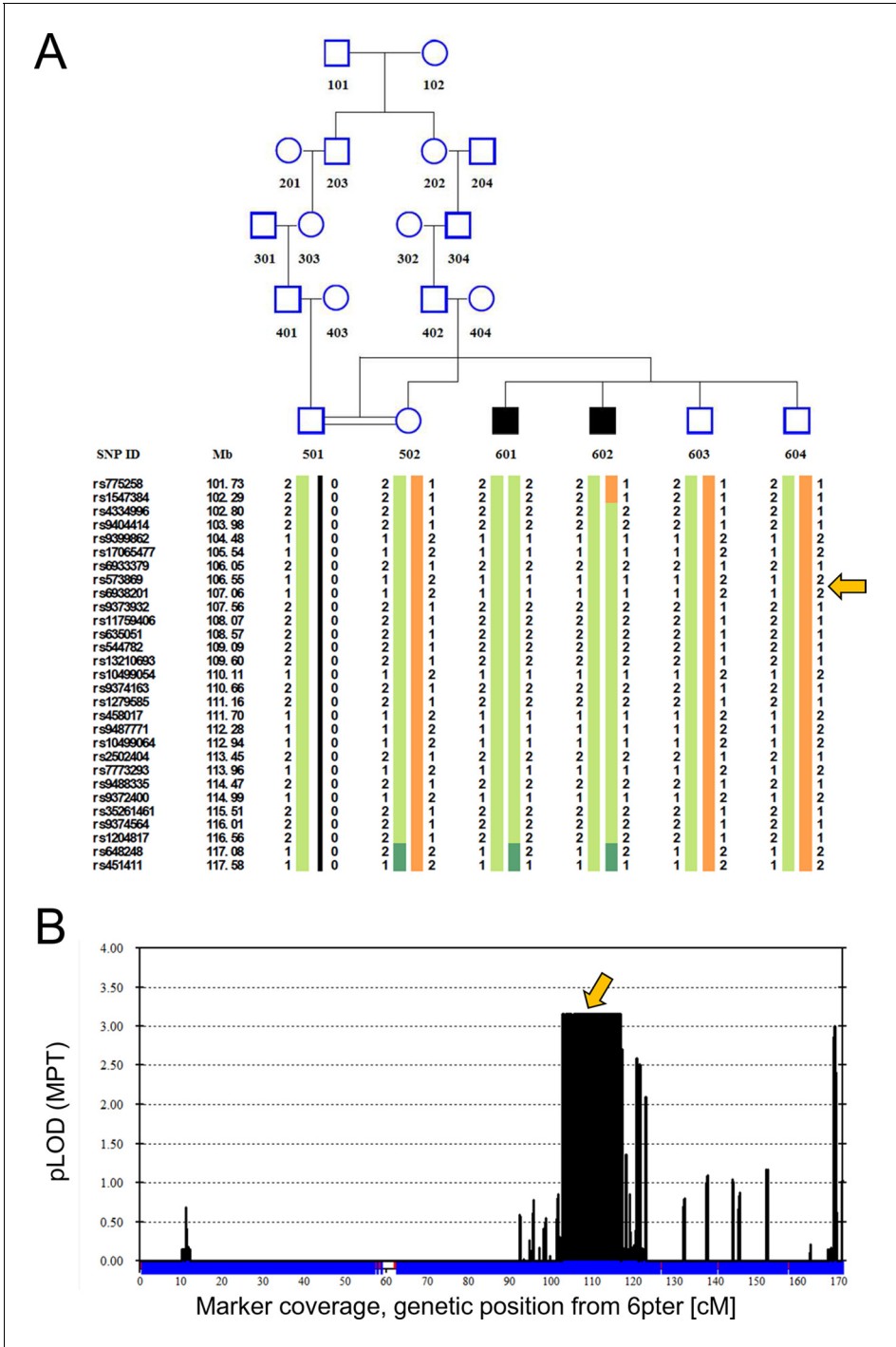

**Figure 1.** Linkage analysis in consanguineous family with two siblings with ataxia, mental retardation and developmental delay maps defect to chromosomal interval containing *ATG5*. Remote consanguinity was detected between parents of two previously described siblings having ataxia (*Yapici and Eraksoy, 2005*), illustrated here as third cousins. SNP and linkage results for chromosome 6 (B) are illustrated below the pedigree (A). The shared homozygous region lies between rs4334996 and rs1204817, encompassing ATG5 at 106.6 Mb. Father (501)'s alleles were inferred, 0 denotes unknown alleles. Affected siblings, 601 and 602, are denoted by black squares and unaffected family members by open symbols. The proximal boundary is defined by a recombination event between rs1547384 and rs4334996 in affected individual 602, while the distal boundary is defined as an ancestral recombination event (lack of homozygosity, dark green) between rs1204817 and rs648248. Orange arrows indicate the position of *ATG5*.

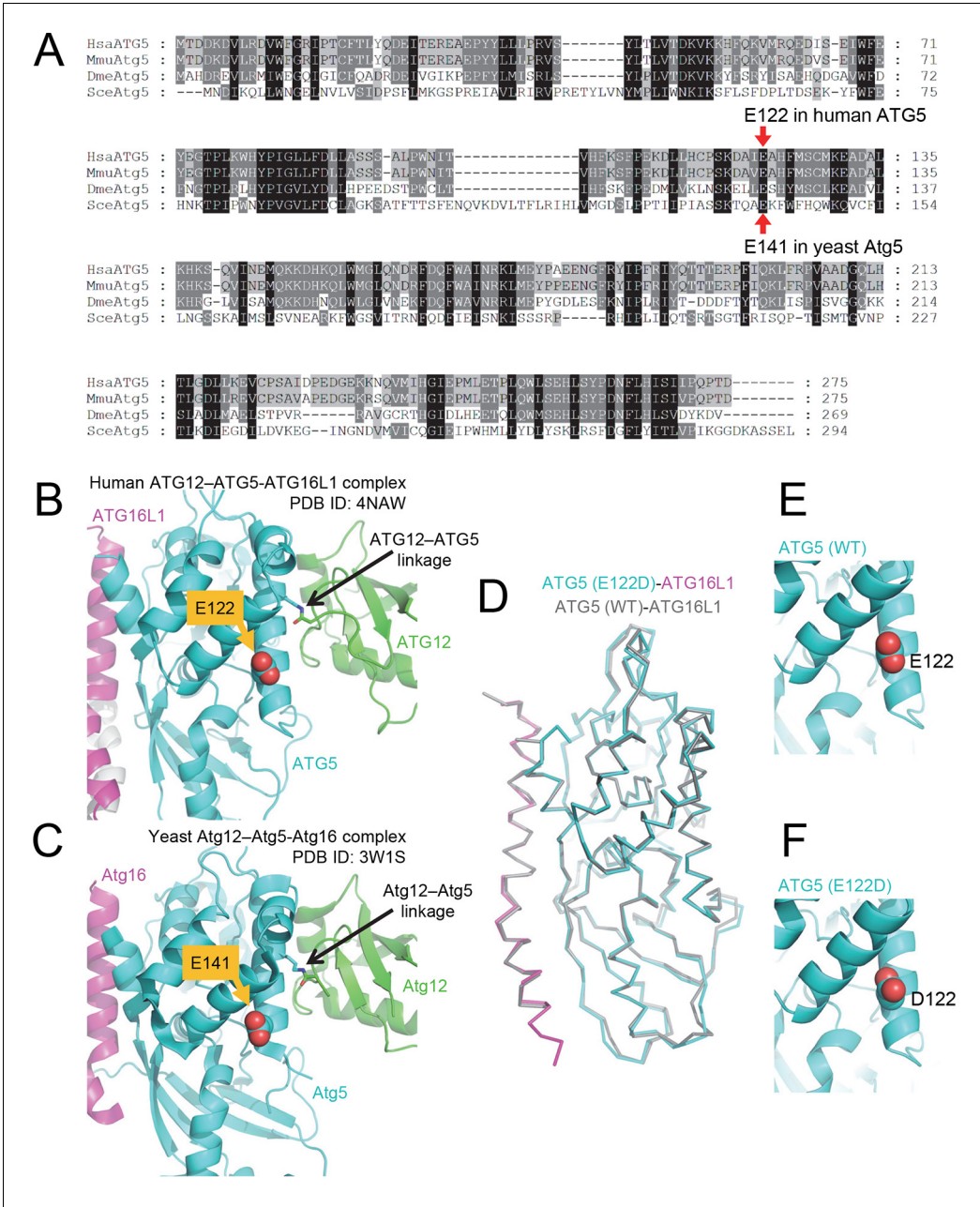

**Figure 2.** The primary sequence of ATG5, including the mutant E122 residue, as well as the protein structure is highly conserved across eukaryotic species. (**A**) Amino acid sequence alignment between ATG5 orthologs from human (HsaATG5), mouse (MmuAtg5), *Drosophila melanogaster* (DmeAtg5) and *Saccharomyces cerevisiae* (SceAtg5) was constructed at GenomeNet (Kyoto University Bioinformatics Center) through CLUSTALW and rendered in Genedoc v.2.7 using default settings. E122 in human ATG5 and E141 in yeast Atg5, which are homologous residues, are indicated by red arrows. (**B**) Location of E122 residue is highlighted in yellow on the crystal structure of a human ATG12 (residues 53–140)–ATG5 -ATG16L1 (residues 11–43) complex (PDB ID: 4NAW). ATG5 is shown in cyan, ATG16L1 in magenta, and ATG12 in green (*Otomo et al., 2013*). (**C**) Location of the E141 residue in yeast Atg5, which corresponds to the E122 in human ATG5, is indicated in yellow on the crystal structure of a yeast Atg12 (100–186)–Atg5 -Atg16 (1–46) complex, colored as for the human counterparts as in panel B (PDB ID: 3W1S) (*Noda et al., 2013*). (**D**) Superimposition of crystal structure of ATG5^E122D-ATG16L1 with ATG5^WT-ATG16L1 (PDB: 4TQ0) (*Kim et al., 2015a*). Close-up view of ATG5 structure around WT (**E**) and E122D (**F**) mutation.

## Cells from ATG5[E122D/E122D] patients exhibit reduced ATG12–ATG5 expression

ATG5 plays a role in elongation of the phagophore and its subsequent maturation into the complete autophagosome. ATG12 is a ubiquitin-like protein that covalently binds ATG5 (*Mizushima et al., 1998a*), and this conjugate noncovalently binds ATG16L1. Crystal structure of the resulting ATG12–ATG5-ATG16L1 complex indicated that E122 is located in the vicinity of the ATG12–ATG5 interaction surface (*Figure 2B*); hence, we predicted that the mutation in ATG5 could affect the conjugation of ATG12. Comparison of protein isolated from control lymphoblastoid cell lines (LCL) and of affected subjects revealed a severe reduction of the ATG12–ATG5 conjugate in the mutant cells under basal conditions (*Figure 3A*), suggesting that the E122D mutation may have impaired autophagy by inhibiting conjugation between ATG12 and ATG5.

## Cells from ATG5[E122D/E122D] patients exhibit autophagic flux attenuation

The ATG12–ATG5-ATG16L1 complex functions in part as an E3 ligase to facilitate the conjugation of LC3 to phosphatidylethanolamine, generating LC3-II (*Fujita et al., 2008*; *Hanada et al., 2007*). Compared to control cells, LCLs from patients with the E122D mutation exposed to bafilomycin $A_1$ showed a substantial reduction in LC3-II accumulation under basal conditions (*Figure 3B*), suggesting a possible decrease in E3 activity and subsequent attenuation of basal autophagic flux. The patient LCLs were also unable to upregulate their autophagic flux in response to Torin 1 (*Figure 3C*), which is a strong inducer of autophagy (*Thoreen et al., 2009*). ATG5[E122D] LCLs also showed elevated levels of SQSTM1/p62, an autophagy receptor and substrate, further indicating disruption of basal autophagy (*Figure 3C*).

## E122D mutation of ATG5 impairs ATG12–ATG5 conjugation

To examine the effect of the ATG5[E122D] mutation on formation of the ATG12–ATG5-ATG16L1 complex, we expressed the recombinant human proteins in insect Hi5 cells and analyzed the complexes by affinity isolation. We could detect the ATG12–ATG5 complex when both wild-type proteins were co-expressed, but we could only detect a minimal amount of the ATG12–ATG5[E122D] complex (*Figure 4A*). Although overexpression of human ATG5[WT] in HEK293 cells or *Drosophila* tissues resulted in efficient covalent conjugation with overexpressed human ATG12 (*Figure 4B and C*) or endogenous *Drosophila* Atg12 (*Figure 4D*), mutant ATG5[E122D] was dramatically impaired in this process (*Figure 4B, C and D*). Interestingly, expression levels of ATG5[WT] and ATG5[E122D] monomers were comparable to each other, indicating that the mutation affects the conjugation process, rather than the stability of proteins. This was consistent with the structural location of ATG5 E122 adjacent to the surface that interacts with ATG12 (*Figure 2B*). To confirm that the mutation does not overtly alter the structure of ATG5 or binding to ATG16L1, we analyzed formation of the noncovalent ATG5-ATG16L1 complex using constructs containing a TEV protease site. Both wild-type and mutant ATG5 protein were efficiently co-precipitated with ATG16L1 (*Figure 4E*). Indeed, the co-crystal structure of a human ATG5[E122D]-ATG16L1 complex (*Figure 2* and *Table 1*) superimposes well with the previously determined structure of the WT proteins (*Figure 2D*), with the major obvious difference being replacement of the side-chain (*Figure 2E and F*). Thus, it appears that the E122D mutation interferes with the ATG12–ATG5 conjugation process, but not with ATG5 folding or binding of ATG16L1.

## ATG5 mutation in yeast results in decreased autophagy

ATG5 is a highly conserved protein, and sequence alignment demonstrated that E122 corresponds to yeast E141 (*Figure 2A and C*). We extended our analysis of the effect of the mutation on autophagy activity, by taking advantage of the yeast system. To test whether autophagy was affected by the *Atg5* mutation in yeast, we initially relied on the GFP-Atg8 processing assay (*Shintani and Klionsky, 2004*). During autophagy a population of Atg8 is continuously transported to the vacuole inside of autophagosomes. Tagging the N terminus of Atg8 with GFP makes it possible to monitor autophagy flux because Atg8 is rapidly degraded inside the vacuole whereas GFP is relatively resistant to vacuolar hydrolases; the generation of free GFP is an indication of autophagic activity. We observed a consistent decrease in autophagy activity with the Atg5[E141D] mutant relative to Atg5[WT] following autophagy induction by starvation (*Figure 5A*). Atg8, or GFP-

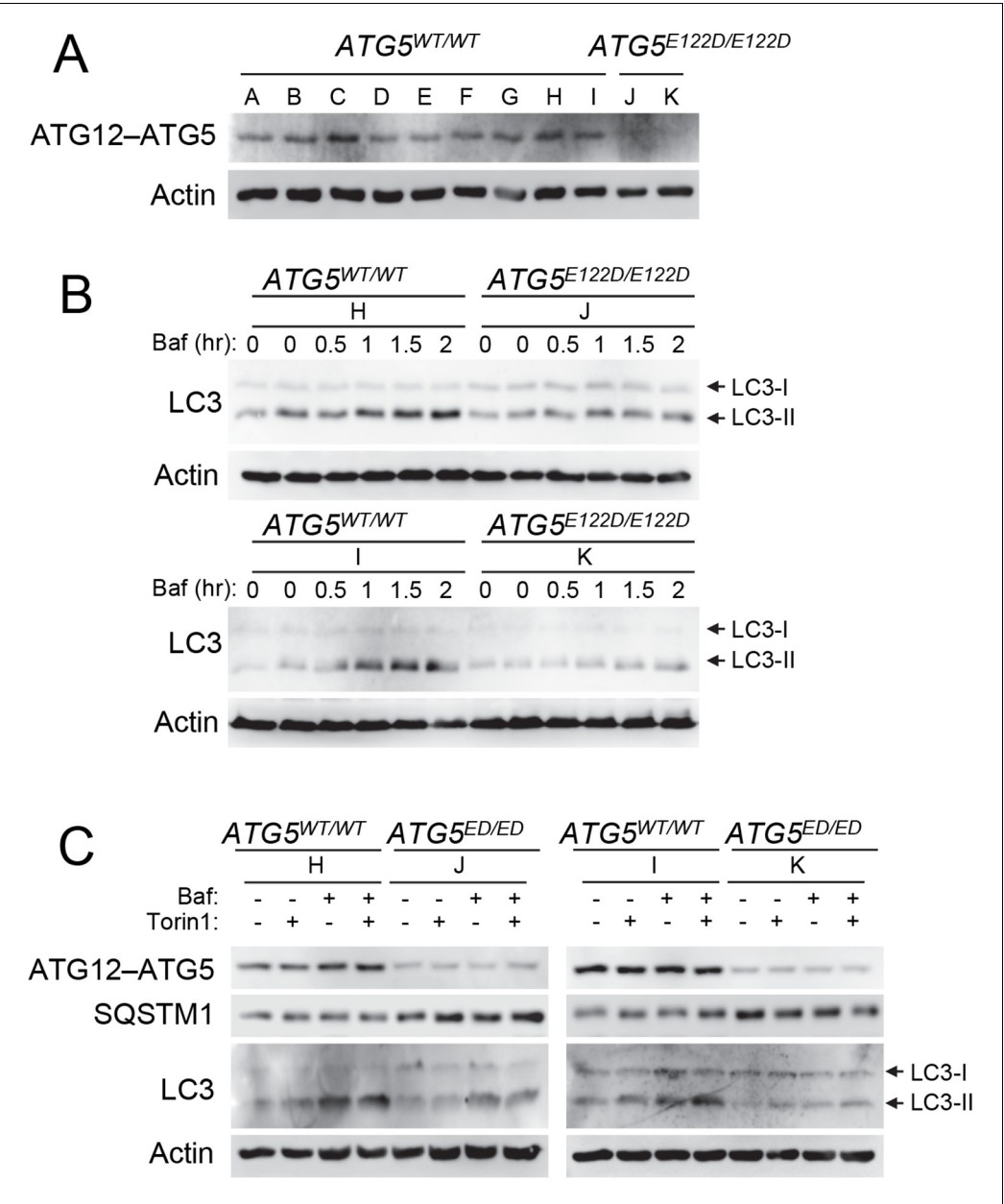

**Figure 3.** Cells from ataxia patients with *ATG5E122D/E122D* mutation exhibit autophagy defects. (**A**) Decreased expression of ATG12–ATG5 conjugates in cells from ataxia patients with *ATG5E122D/E122D* mutation. ATG5 immunoblotting (IB) of ATG12–ATG5 conjugates of LCLs from individuals whose *ATG5* genotype corresponds to wild type (A to I) or E122D (J and K). (**B**) Decreased autophagic flux in *ATG5E122D/E122D* LCL cells. A subset of LCLs from (**A**) were treated with 0.1 μM bafilomycin $A_1$ (Baf) for the indicated hours and analyzed by IB. LC3-II is an autophagosome marker, and LC3-I is a precursor for LC3-II. Baf inhibits lysosomal degradation of LC3-II. Actin is shown as a loading control. (**C**) Decreased autophagic flux and increased expression of SQSTM1, an autophagy substrate, in *ATG5E122D/E122D* LCL cells. A subset of LCLs from (**A**) were treated with 250 nM Torin 1 or 0.1 μM Baf, for 2 hr and analyzed by IB. Torin 1 is an autophagic flux activator.

Atg8, does not measure autophagic cargo per se (*Klionsky, 2016*), and the amount of GFP-Atg8 processing only corresponds to the inner surface of the autophagosome. To corroborate the effects observed through the GFP-Atg8 processing assay we examined autophagy using the quantitative Pho8Δ60 assay (*Noda and Klionsky, 2008*). Pho8Δ60 is an altered form of a phosphatase

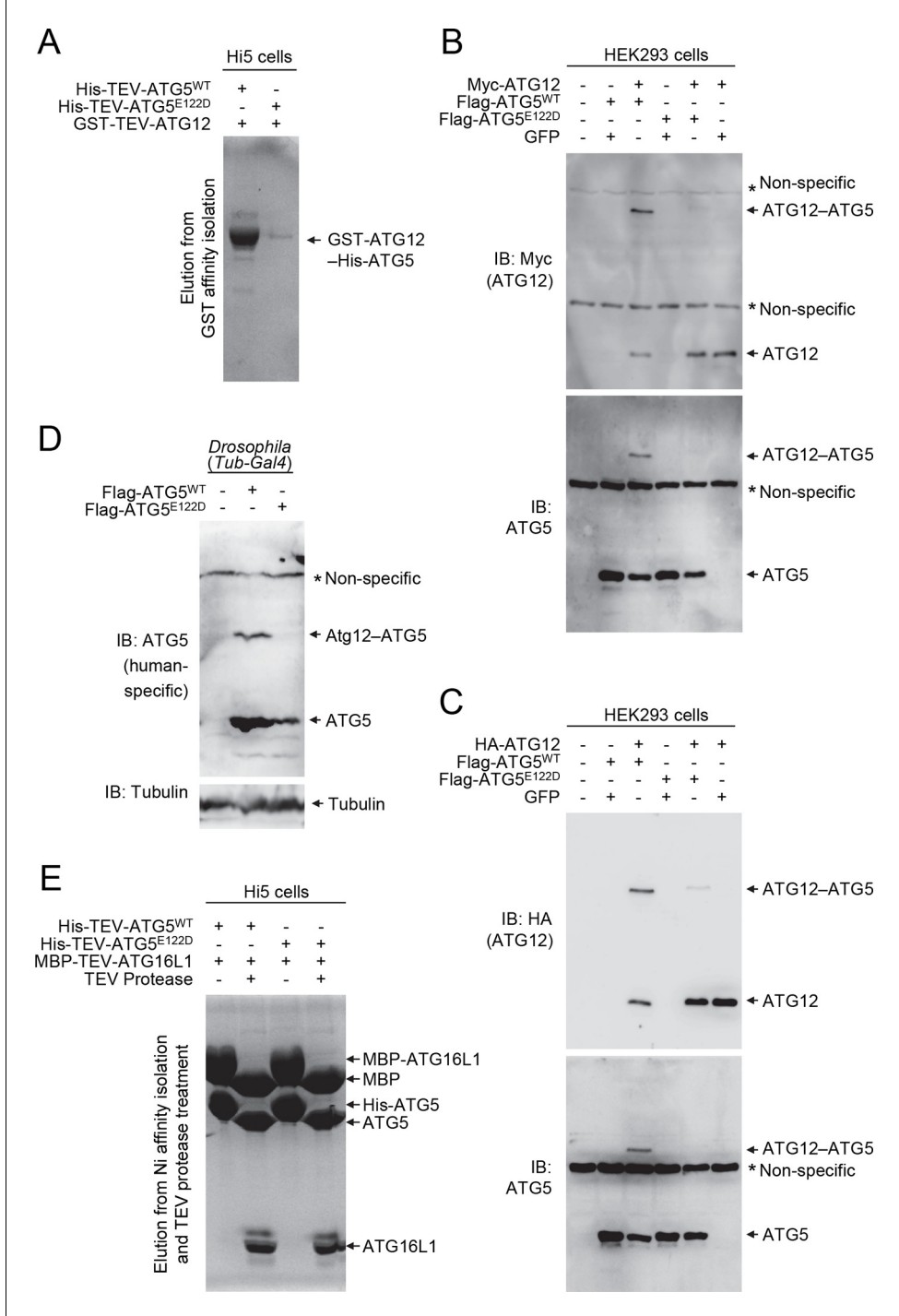

**Figure 4.** E122D mutation interferes with formation of the ATG12–ATG5 conjugate. (**A**) Coomassie Blue-stained SDS-PAGE gel following glutathione affinity purification from lysates of Hi5 cells infected with baculoviruses expressing GST-ATG12 and either WT or E122D mutant ATG5. (**B** and **C**) HEK293 cells expressing the indicated proteins were analyzed by IB. (**D**) *Drosophila* whole bodies expressing the indicated transgenes under the control of *Tub-Gal4* were analyzed by IB. (**E**) Lysates from Hi5 cells expressing the indicated proteins were subjected to His/Ni-NTA purification and subsequent TEV protease treatment. Proteins were analyzed by Coomassie Blue staining.

**Table 1.** Crystallography data collection and refinement statistics.

| Data collection | |
|---|---|
| Beam line | APS 24-ID-C |
| Space group | C2 |
| Unit cell parameters | |
| a, b, c (Å) | 217.1, 84.5, 151.9 |
| α, β, γ (°) | 90, 133.8, 90 |
| Resolution (Å) (highest shell) | 50–3.0 (3.12-3.0) |
| Wavelength (Å) | 0.9792 |
| Number of measured reflections | 179,310 |
| Number of unique refections | 39,496 |
| Overall $R_{sym}$ | 0.057 (0.645) |
| Completeness (%) | 98.9 (99.3) |
| Overall $I/\sigma I$ | 14.3 (2.3) |
| Multiplicity | 4.5 |
| Refinement | |
| Resolution (Å) | 50–3.0 |
| $R_{work}/R_{free}$ | 0.198/0.244 |
| rmsd bond lengths (Å) | 0.008 |
| rmsd bond angles (°) | 0.994 |
| Number of protein atoms | 9403 |
| Ramachandran statistics | |
| Preferred (%) | 97.69 |
| Allowed (%) | 2.22 |
| Disallowed (%) | 0.09 |

that is only delivered to the vacuole via autophagy; subsequent proteolytic processing generates an active form of the hydrolase. After 4 and 6 hr of starvation, yeast cells expressing the plasmid-based Atg5$^{E141D}$ mutant showed a significant decrease in autophagy levels compared to cells expressing Atg5$^{WT}$ (*Figure 5B*), and similar results were obtained when the WT and mutant *ATG5* genes were integrated back into the chromosomal *ATG5* locus (*Figure 5C*).

To determine the reason for reduced autophagic activity we tested the effects of the Atg5$^{E141D}$ mutant on Atg8 lipidation. As shown by the ratio of Atg8–PE:total Atg8, cells expressing Atg5$^{E141D}$ displayed a decrease in Atg8–PE conjugation at 30 and 60 min of starvation compared to cells expressing Atg5$^{WT}$ (*Figure 5D*). We extended this analysis using the in vivo reconstitution of Atg8–PE conjugation as described previously (*Cao et al., 2008*). In brief, we examined Atg8 lipidation in a multiple-knockout (MKO) strain in which 23 *ATG* genes are deleted, when expressing only the E1, E2 and E3-like conjugation enzymes of the autophagy machinery. Atg8ΔR that lacks the C-terminal arginine was used in the assay to bypass the initial activation step initiated by Atg4; due to the absence of Atg4, there is no cleavage of Atg8–PE from the membrane, resulting in stabilization of this form of the protein. We found that the Atg5$^{E141D}$ mutant was significantly defective in Atg8–PE conjugation compared to the cells with Atg5$^{WT}$ when Atg16 was not present (*Figure 5E*). Atg16 is not required mechanistically for Atg8 conjugation, but its presence increases the efficiency of this process and may dictate the site of conjugation (*Cao et al., 2008*; *Hanada et al., 2007*). Thus, the presence of Atg16 may partially mask the Atg8 lipidation defects of the Atg5$^{E141D}$ mutant, and this may explain why the E122D/E141D mutation induces a hypomorphic rather than a complete null phenotype.

## ATG5$^{E122D}$ fails to complement the ataxic phenotype of *Atg5*-null flies

To further characterize the effect of the E122D mutation on the development of ataxia, we generated *Drosophila melanogaster* knockouts for *Atg5* (*Figure 6A*), and reconstituted the *Atg5*-null mutant flies with transgenes expressing wild-type (WT) or E122D human ATG5 (*Figure 6B-D*). Unlike mouse models, *Atg5*-null flies are viable, although they exhibit severe mobility defects after adult eclosion as demonstrated by a negative geotaxis assay (*Figure 6E and I*, and *Video 1*), similar to *Atg7* null mutant flies (*Juhasz et al., 2007*). These mobility defects were substantially restored by expression of ATG5$^{WT}$ (*Figure 6F and I*, and *Video 2*), suggesting that the molecular function of ATG5 is conserved between human and *Drosophila*. However, *Atg5*-null mutant flies expressing ATG5$^{E122D}$ were still defective in mobility although slightly better than *Atg5*-null controls (*Figure 6G–I*, and *Videos 3* and *4*), demonstrating again that ATG5 activity is compromised but not eliminated by the E122D mutation. ATG5$^{E122D}$ was also inferior to ATG5$^{WT}$ in suppressing Ref(2)P (fly p62/SQSTM1) accumulation (*Figure 6J and K*) and cell death (*Figure 6L and M*) in the brain of *Atg5*-null mutant flies.

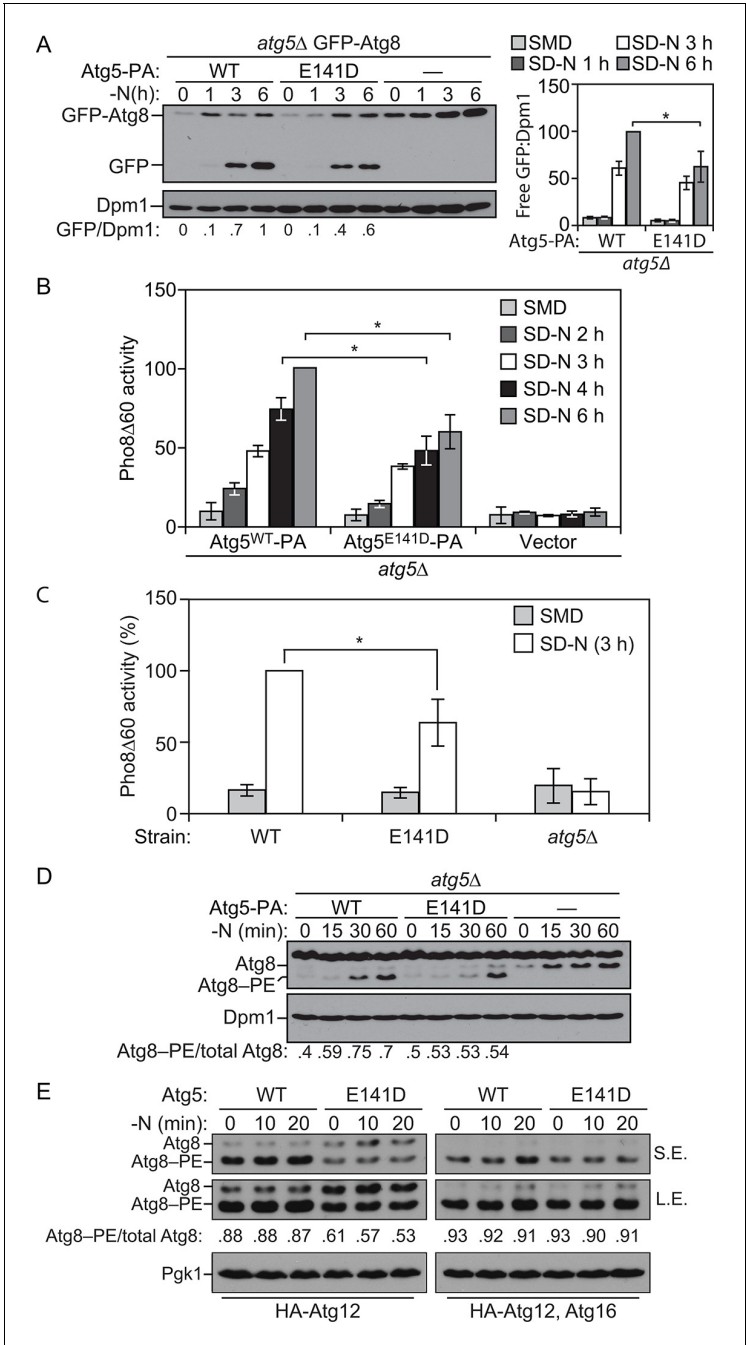

**Figure 5.** E141D mutation of yeast Atg5 attenuates autophagy. (A-D) Yeast cells were grown in SMD to mid-log phase and nitrogen starved for the indicated times. (A) WLY176 *atg5Δ* yeast cells expressed GFP-Atg8 through its endogenous promoter and plasmid-based Atg5[WT]-PA, Atg5[E141D]-PA or an empty vector. Protein extracts were analyzed for GFP-Atg8 processing by western blot. The ratio of free GFP to Dpm1 (loading control) is presented below the blots, and quantification is presented on the right (Student's t test, n=4; *p < 0.05); the value for Atg5[WT] at 6 hr was set to 1.0 and other values were normalized. (B) WLY176 *atg5Δ* yeast cells expressed either plasmid-based Atg5[WT]-PA, Atg5[E141D]-PA or an empty vector. Protein extracts were used to measure autophagy through the Pho8Δ60 assay (Student's t test, n=6; *p < 0.05). (C) WLY176 cells with genomic integrated Atg5[WT] or Atg5[E141D] were used to generate protein extracts and autophagy was monitored through the Pho8Δ60 assay (Student's t test, n=3; *p < 0.05). (D) WLY176 *atg5Δ* yeast cells expressing plasmid-based Atg5[WT]-PA, Atg5[E141D]-PA or an empty vector were used to generate protein extracts. The ratio of Atg8–PE to total Atg8 is presented below the blots based on western blot using antiserum to Atg8. Dpm1 was used as a loading control. (E) *MKO ATG3* (YCY137) cells were co-transformed with pATG8ΔR-ATG7-ATG10(414), and either pATG5[WT]-HA-ATG12(416), pATG5[E141D]-HA-ATG12(416), pATG5[WT]-HA-ATG12-ATG16(416), or pATG5[E141D]-HA-ATG12-ATG16(416). Overnight cultures were diluted to OD=0.02 in SMD -Ura -Trp. The cells were incubated at 30℃ for 18 hr to mid-log phase before they were shifted to SD-N for nitrogen starvation. Samples at the corresponding time points were collected, TCA precipitated and subsequently analyzed by western blot. S.E., short exposure; L.E., long exposure.

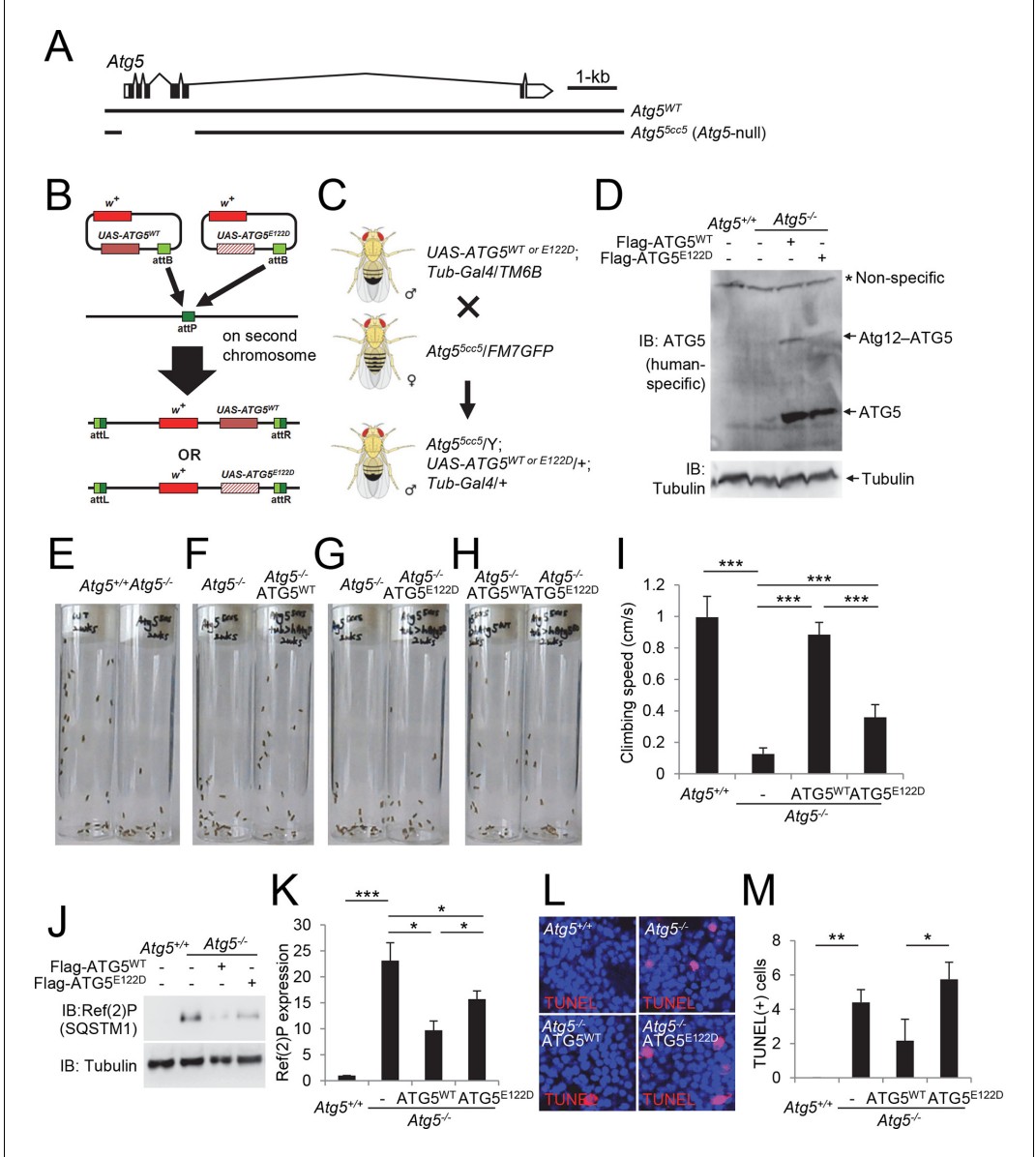

**Figure 6.** Ataxic phenotype of *Atg5*-null flies is suppressed by human ATG5[WT] but not by ATG5[E122D]. (**A**) Genomic organization of the *Atg5* locus and the *Atg5*-null mutant (*Atg5[5cc5]*). *Atg5[5cc5]* mutants have a CRISPR-Cas9-mediated deletion in approximately 1.5 kb residues that eliminate more than 85% of Atg5-coding sequences including the translation start site. Open boxes, untranslated exons; closed boxes, protein-coding exons. Scale bar, relative length of 1 kb genomic span. (**B**) Schematic representation of how *ATG5* transgenic flies were made. Plasmid which can express wild-type or E122D-mutated human *ATG5* was inserted into an identical genomic location (the attP site) through phiC31-mediated recombination (*Bateman et al., 2006*; *Bischof et al., 2007*; *Venken et al., 2006*). The scheme was adapted from a previous publication (*Kim and Lee, 2015*). (**C**) Genetic scheme of how *ATG5* transgenes were placed into the *Atg5*-null mutant flies. *Atg5, UAS-ATG5* and *Tub-Gal4* loci are on the X-chromosome, second chromosome and third chromosome, respectively. (**D**) Whole flies of indicated genotypes were analyzed by IB. (**E** to **H**) Photographs of the vials containing 2-week-old adult male flies of indicated genotypes taken at 3 sec after negative geotaxis induction: (**E**) *Atg5*-null flies exhibit severely impaired mobility. (**F**) Ataxic phenotype of *Atg5*-null flies is complemented by human ATG5[WT] expression. (**G** and **H**) Human ATG5[E122D] is less capable than human ATG5[WT] in suppressing the fly ataxia phenotype. (**I**) Quantification of the climbing speeds of 2-week-old adult male flies (n≥20) of the indicated genotype. Climbing speed is presented as mean ± standard deviation (n=5). P values were calculated using the Student's t test (***p<0.001). (**J**) *Drosophila* heads from two-weeks-old flies of the indicated genotypes were analyzed by IB. (**K**) Ref(2)P [p62] is an autophagy substrate. Relative protein expression was measured by densitometry and presented in a bar graph (mean ± standard error; n=4). (**L**) Terminal deoxynucleotidyl transferase dUTP nick end labeling (TUNEL) of *Drosophila* brain (middle layer of the medial compartment). (**M**) TUNEL-positive cells per field were quantified and presented in a bar graph (mean ± standard error; n≥5). K and M: *P* values were calculated using the Student's t test (*p<0.05, **p<0.01, ***p<0.001).

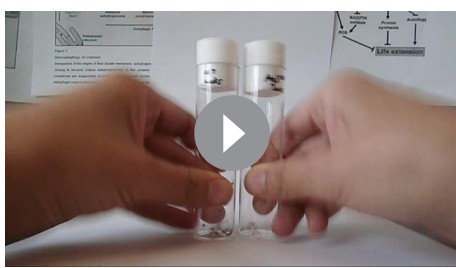

**Video 1.** Climbing assay in 2 weeks-old wild-type flies (left) and *Atg5*-null flies (right).

**Video 2.** Climbing assay in 2 weeks-old *Atg5*-null flies (left) and *Atg5*-null flies expressing *ATG5*^WT (right).

## Discussion

In summary, we demonstrate that the homozygous E122D mutation of ATG5, a unique mutation found in two human subjects with ataxia, results in reduced conjugation to ATG12 and in an overall decrease in autophagy activity. The homologous mutation in yeast also interferes with autophagy, and the ataxia phenotype was replicated in a fly model. Based on these results we propose that this *ATG5* mutation, and the consequent disruption in autophagy activity, is the cause of the ataxic phenotype and disturbance of the cerebellum in the affected siblings. This hypothesis is in agreement with previously characterized mouse models, in which neuron-specific knockout of *Atg5* results in ataxia-like phenotypes (*Hara et al., 2006*; *Nishiyama et al., 2007*). By contrast, mice with complete knockout of *Atg5* die shortly after birth, demonstrating that autophagy is essential for mammalian survival (*Kuma et al., 2004*). Our results indicate that E122D is a partial loss-of-function allele that impairs but does not completely abolish ATG5 activity. Although the overall structure of the E122D mutant ATG5 superimposes well with the wild-type protein, the mutation causes a striking decrease in the level of ATG12–ATG5 conjugate that is formed when the C terminus of ATG12 is covalently linked to Lys130 of ATG5. We speculate that the E122D mutation causes subtle changes in the conformational dynamics that propagate to Lys130, which is less than 10 Å away, resulting in less ATG12–ATG5, which in turn leads to reduced LC3/Atg8 conjugation.

In neurons, which are among the cells most dependent on autophagy for tissue homeostasis (*Button et al., 2015*), the residual function of the E122D allele is inadequate, resulting in predominantly neurological symptoms in the two patients. Since homozygous mutations with complete loss-of-function have not been reported, we predict that individuals carrying such mutations, similar to *Atg5*-null mice (*Kuma et al., 2004*), might not be viable.

Autophagy is quickly gaining importance for its roles in preventing neurodegeneration. *WDR45* is a redundant, non-core autophagy gene, one of four mammalian homologs to Atg18, and mutations in *WDR45* cause SENDA, static encephalopathy of childhood with neurodegeneration in adulthood (*Haack et al., 2012*; *Saitsu et al., 2013*). Autophagy appears to be critical in ataxia, whether mutant proteins evade autophagy processes or normal autophagy is disrupted. Several ataxias are attributed to intranuclear or cytoplasmic aggregation of mutant proteins within the cell (*Matilla-Duenas et al., 2014*). These protein aggregates, in humans and in mouse models, not only evade

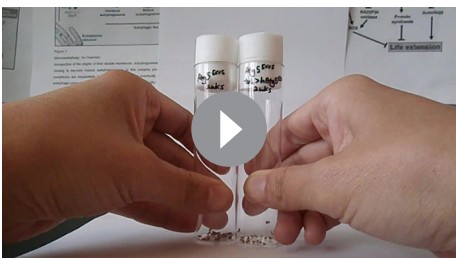

**Video 3.** Climbing assay in 2 weeks-old *Atg5*-null flies (left) and *Atg5*-null flies expressing *ATG5*^E122D (right).

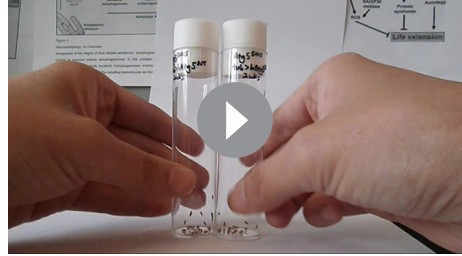

**Video 4.** Climbing assay in 2 weeks-old *Atg5*-null flies expressing *ATG5*^WT (left) or *ATG5*^E122D (right).

autophagic sequestration but may even inhibit autophagy (*Alves et al., 2014*), or lead to reduction in autophagy available for other proteins due to saturation. Assessment of autophagy in patient cells may be used to refine and identify the genetic cause of a patient's ataxia.

Further discovery of the role of autophagy in neurodegenerative diseases should be used to investigate therapies targeted at the autophagy process. Many drugs enhance autophagy and their effects on a multitude of neurodegenerative diseases have been studied (*Sarkar et al., 2009*). Recently more studies have been conducted assessing the value of autophagy enhancers in ataxia models and patients. Induction of autophagy through administration of Temsirolimus, a rapamycin ester, and lentiviral overexpression of BECN1 in SCA3 model mice increase autophagy and the clearance of mutant protein aggregates, and reduce the ataxic phenotype (*Menzies et al., 2010*; *Nascimento-Ferreira et al., 2013*). In a single patient trial, trehalose treatment of patient fibroblasts increased autophagy and alleviated cellular pathogenic features by improving mitochondrial morphology and reducing free radicals in the cell (*Casarejos et al., 2014*; *Sarkar et al., 2007*). Trehalose also showed success in trials involving models of SCA17 (*Chen et al., 2015*). Lithium, another inducer of autophagy, improved symptoms in a SCA1 mouse model (*Watase et al., 2007*), but did not slow or reduce symptoms in a treatment trial in SCA2 patients (*Sacca et al., 2015*). An autophagy enhancer may be an appropriate treatment to test in the presented subjects, as autophagic flux is attenuated, but not completely abrogated, by the $ATG5^{E122D/E122D}$ mutation.

This study's finding of the pathogenic human E122D mutation in *ATG5*, a gene encoding part of the autophagy-controlling core machinery, is important and novel, but consistent with reports of neurodegenerative disorders in other autophagy-related genes (*Frake et al., 2015*). Our results suggest that other mutations in this and other *ATG* genes, which impair but do not completely abolish autophagy, may result in similar forms of ataxia, intellectual disability and developmental delay. This study exemplifies the utility of exome sequencing in the identification of rare disease-causing variants, and supports the role of impaired autophagy in neurodegenerative disease. In addition, we demonstrate the utility of a combined genetic, biochemical and cell biological analysis in multiple model systems to elucidate the underlying pathogenic mechanism of rare human diseases.

# Materials and methods

## Subjects

Study protocols including written informed consents have been approved by the University of Michigan Institutional Review Board and the Boğaziçi University Institutional Review Board for Research with Human Participants. Two Turkish brothers, ages 5 and 7 in 2004, presented with ataxia and developmental delay, as previously described (*Yapici and Eraksoy, 2005*). Parents were initially reported to be unrelated, but recently suggested they might be remotely related. Both patients were delayed in walking, had truncal ataxia and dysmetria, nystagmus, and lower IQ (68 and 70). MRI revealed cerebellar hypoplasia. Follow-up examinations showed no progression of symptoms.

## Genetic analysis

DNA was isolated from peripheral whole blood using the Qiagen (Germantown, MD) Gentra Puregene isolation kit. Linkage analysis was performed using the genotype data generated with Illumina HumanOmniExpress-24 chip for the mother and the four sibs. The Allegro module (*Gudbjartsson et al., 2000*) of easyLINKAGE software was used, assuming autosomal recessive inheritance and parents as third cousins. No deletion or duplications common to just the two affected brothers were detected using cnvPartition plug-in in Illumina Genome Studio v.1.02 software.

Exome sequencing was performed independently twice on one subject. Capture for whole exome sequencing was performed with NimbleGen SeqCap EZ Exome Library v1.0 kit (Roche, Indianapolis, IN). Captured regions were sequenced with Illumina HiSeq2000 instruments. Variants were filtered to remove common variants based on 1000 Genomes, Exome Sequencing Project, and Exome Aggregation Consortium databases, variants outside of identified linkage regions, variants not expected to change protein coding, and variants not following a recessive model of inheritance (*Exome Aggregation Consortium (ExAC), 2015*; *Genomes Project Consortium et al., 2012*; *NHLBI Go Exome Sequencing Project, 2015*).

PCR followed by Sanger sequencing was performed to validate the variant identified through exome sequencing and test for segregation within the family. The variant of interest was further examined in two separate collections of a total of 500 Turkish samples, and found absent.

## Lymphoblast cell culture

Lymphoblastoid cell lines (LCL) of both subjects were generated from heparinized whole blood samples and cultured as described (*Doyle, 1990*). As they are made in house and cultured briefly, mycoplasma contamination risk is minimized.

## Protein co-expression and affinity purification from insect cells

We used a baculovirus/insect cell expression system to examine formation of the human ATG12–ATG5 conjugate in a heterologous system described previously (*Qiu et al., 2013*). Hi5 insect cells (Invitrogen, Carlsbad, CA) were infected with baculoviruses expressing human ATG7, ATG10, a GST-tagged version of ATG12 (residues 53–140, corresponding to the ubiquitin-like domain) and a His-tagged version of either $ATG5^{WT}$ or $ATG5^{E122D}$. Three days post infection, lysates were subjected to glutathione affinity chromatography, and the GST-ATG12–His-ATG5 conjugate was detected by SDS-PAGE followed by Coomassie Blue staining. To confirm that the baculoviruses produce protein, Hi5 cells were coinfected with baculoviruses expressing the His-tagged WT and mutant versions of ATG5 and the N-terminal domain of ATG16L1 (residues 1–69, as an MBP fusion). At three days post infection, lysates were subjected to nickel affinity purification. The ATG5-ATG16L1 complex formation was detected by SDS-PAGE and Coomassie Blue staining.

## Crystallization and structure determination

The complex containing $ATG5^{E122D}$ and the N-terminal domain of ATG16L1 (residues 1–69) was expressed in Hi5 insect cells, and purified by nickel affinity, ion exchange, and size exclusion chromatography into a final buffer of 20 mM Tris, pH 8.5, 50 mM NaCl, 10 mM DTT. The complex was concentrated to 18.5 mg/ml, aliquoted, flash-frozen and stored at -80℃ until further use. Crystals were grown by the hanging drop vapor diffusion method by mixing purified protein 1:1 with reservoir solutions of 37.5 mM MES, pH 5.2–5.8, 0.2 M sodium tartrate, and 11–13% polyethylene glycol 3350. Final crystals were obtained by micro-seeding with reservoir solution of 40 mM MES, pH 5.5, 0.2 M sodium tartrate, 8.5% PEG3350, 10 mM DTT. Crystals were cryoprotected in reservoir solution supplemented with 25% xylitol, and flash frozen in liquid nitrogen prior to data collection. Diffraction data were processed with XDS. The structure was determined by molecular replacement using Phaser (*McCoy et al., 2007*) with the structure of the WT ATG5-ATG16L1 (1–69) (PDB: 4TQ0) complex as a search model (*Kim et al., 2015a*). Model construction and rebuilding were performed using Coot (*Emsley et al., 2010*). The structure was refined using Phenix (*Adams et al., 2010*). Diffraction data and refinement statistics are provided in *Table 1*.

## Immunoblotting

Cells or tissues were lysed in cell lysis buffer (20 mM Tris-HCl, pH 7.5, 150 mM NaCl, 1 mM EDTA, 1 mM EGTA, 2.5 mM sodium pyrophosphate, 1 mM beta-glycerophosphate, 1 mM $Na_3VO_4$, 1% Triton X-100) or RIPA buffer (50 mM Tris-HCl, pH 7.4, 150 mM NaCl, 1% sodium deoxycholate, 1% NP-40, 0.1% SDS) containing protease inhibitor cocktail (Roche). After being clarified with centrifugation, lysates were boiled in SDS sample buffer, separated by SDS-PAGE, transferred to polyvinylidene difluoride membranes and probed with the indicated antibodies. ATG5 (12994), LC3 (3868) and SQSTM1/p62 (5114) antibodies were purchased from Cell Signaling Technology. Hemagglutinin (HA, 3F10) antibody was from Roche. Actin (JLA20) and tubulin (T5168) antibodies were from Developmental Studies Hybridoma Bank and Sigma, respectively. Ref(2)P antibody was previously described (*Pircs et al., 2012*).

## HEK293 cell culture

Wild-type human ATG5-coding sequence was from Addgene #24922 (deposited by Dr. Toren Finkel) (*Lee et al., 2008*). The E122D mutation was introduced into *ATG5* by PCR-based site-directed mutagenesis. $ATG5^{WT}$ and $ATG5^{E122D}$ were cloned into the plasmid pLU-CMV-Flag. The HA-ATG12-expressing plasmid was from Addgene #22950 (deposited by Dr. Noboru Mizushima)

(*Mizushima et al., 1998b*). HEK293 cells (the 293 A substrain from Invitrogen, tested negative for mycoplasma by PCR) were cultured in Dulbecco's modified Eagle's medium (DMEM, Invitrogen) containing 10% fetal bovine serum (FBS) and penicillin/streptomycin at 37°C in 5% CO2. For transient expression of proteins, HEK293 cells were transfected with purified plasmid constructs and polyethylenimine (PEI, Sigma) as previously described (*Horbinski et al., 2001*). Cells were harvested 24 hr after transfection for immunoblot analyses.

## Yeast model

*Saccharomyces cerevisiae* strain WLY176 was used to generate an *ATG5* knockout strain (*atg5Δ*) as previously described (*Gueldener et al., 2002*). The MKO strain YCY137 (SEY6210 *atg1Δ, 2Δ, 4Δ, 5Δ, 6Δ, 7Δ, 8Δ, 9Δ, 10Δ, 11Δ, 12Δ, 13Δ, 14Δ, 16Δ, 17Δ, 18Δ, 19Δ, 20Δ, 21Δ, 23Δ, 24Δ, 27Δ, 29Δ*) (*Cao et al., 2008*), was used for in vivo reconstitution of Atg8 conjugation. Site-directed mutagenesis was performed to generate *ATG5* amplicons with the E141D mutation as previously described (*Liu and Naismith, 2008*). A pRS406 empty plasmid was digested with SpeI and SalI, and then ligated with a DNA fragment encoding either wild-type or mutant Atg5-PA. *atg5Δ* was transformed with an empty pRS406 vector, or plasmids encoding Atg5-PA WT or Atg5-PA E141D. Wild-type WLY176 colonies were transformed with empty pRS406 vector as a control. Colonies were grown on SMD-URA medium and starved in nitrogen-deficient medium. Pho8Δ60 and western blot analyses were performed as described previously (*Noda and Klionsky, 2008*; *Shintani and Klionsky, 2004*). Quantification was performed using ImageJ software. The pATG8ΔR-ATG7-ATG10(414), pATG5 (WT)-HA-ATG12(416) and pATG5(WT)-HA-ATG12-ATG16(416) plasmids were described previously (*Cao et al., 2008*). The pATG5(E141D)-HA-ATG12(416) and pATG5(E141D)-HA-ATG12-ATG16(416) plasmids were made by site-directed mutagenesis based on the wild-type constructs.

## Drosophila genetics

*Atg5*-null Drosophila flies (*Atg5^5cc5^*) were generated by CRISPR-Cas9-mediated genome editing, using a double gRNA approach, both targeting the same gene, as described (*Kondo and Ueda, 2013*). The *Atg5^5cc5^* mutant was recovered by screening viable candidate lines for accumulation of the specific autophagy cargo Ref(2)P using western blots, followed by PCR and sequencing. *Atg5^5cc5^* mutants have a deletion in X:7,322,242–7,323,717 residues (*Drosophila melanogaster* R6.06), which deletes five out of six exons of the *Atg5* gene, eliminating more than 85% of protein-coding sequences including the translation start site (*Figure 6A*). The PhiC31 integrase-mediated site-specific transformation method was used to express human $ATG5^{WT}$ and $ATG5^{E122D}$ from an identical genomic locus (*Bateman et al., 2006*; *Bischof et al., 2007*; *Venken et al., 2006*). In brief, flag-tagged $ATG5^{WT}$ and $ATG5^{E122D}$ were cloned into a pUAST-attB vector (*Bischof et al., 2007*) and fully sequenced. pUAST-attB-$ATG5^{WT}$ and pUAST-attB-$ATG5^{E122D}$ were microinjected into $y^1$ M{vas-int. Dm}ZH-2A w*; M{3xP3-RFP.attP}ZH-51D flies and stable transformants were isolated by the presence of the *mini-white^+* marker (*Figure 6B*). The UAS-$ATG5^{WT}$ or UAS-$ATG5^{E122D}$ transgenes were crossed with a double balancer strain (*Bl/CyO; TM2/TM6B*) and then with *+/CyO; Tub-Gal4/TM2* to be constructed as stable *Tub>ATG5* lines (*UAS-ATG5/UAS-ATG5; Tub-Gal4/TM6B*). The *Tub>ATG5* male flies were crossed with *Atg5^5cc5^/FM7* female flies to generate *Atg5*-null flies expressing human *ATG5* transgenes. Climbing assays and TUNEL staining were performed as previously described (*Kim et al., 2015b*).

## Acknowledgements

This research was funded by the National Institutes of Health (NIH) grants R01-NS078560 (MB), R21-OD018265 (JHL), R01-GM077053 (BAS) and R01-GM053396 (DJK), HHMI (BAS), American Heart Association 14POST19890021 (YQ), the Boğaziçi University Research Fund Grant 6655 (AT), ALSAC/ St. Jude (BAS), the Wellcome Trust 087518/Z/08/Z and Lendulet LP2014-2 (GJ). We thank Miriam H Meisler for critical reading, Karen Majczenko, MD (deceased) and Linda Gates for EBV transformation and tissue culture, Dr. Ke Wang and Kendal Walker for technical assistance, the University of Michigan DNA Sequencing Core for Exome sequencing, Dr. Mustafa Cengiz Yakicier (ACIBADEM University, Turkey, Department of Medical Biology and Genetic Diagnostic Center) for Turkish control samples, and the TUBITAK Advanced Genomics and Bioinformatics Group (IGBAM) for sharing the Turkish exome sequence database.

# Additional information

## Funding

| Funder | Grant reference number | Author |
|---|---|---|
| National Institutes of Health | OD018265 | Myungjin Kim<br>Jun Hee Lee |
| National Institutes of Health | NS078560 | Erin Sandford<br>Jun Z Li<br>Margit Burmeister |
| American Heart Association | 14POST19890021 | Yu Qiu |
| Howard Hughes Medical Institute | | Brenda A Schulman |
| National Institutes of Health | GM077053 | Brenda A Schulman |
| American Lebanese Syrian Associated Charities /St. Jude | | Brenda A Schulman |
| Bogazici University Research Fund | 6655 | Aslıhan Tolun |
| Magyar Tudományos Akadémia | Lendulet LP2014-2 | Gabor Juhasz |
| Wellcome Trust | 087518/Z/08/Z | Gabor Juhasz |
| National Institutes of Health | GM053396 | Daniel J Klionsky |

The funders had no role in study design, data collection and interpretation, or the decision to submit the work for publication.

## Author contributions

MKi, ES, Acquisition of data, Analysis and interpretation of data, Drafting or revising the article; DG, YQ, IS, S-HR, BK, RNM, Acquisition of data, Analysis and interpretation of data; XL, YZ, Acquisition of data, Drafting or revising the article; BAS, AT, JHL, MB, Conception and design, Acquisition of data, Analysis and interpretation of data, Drafting or revising the article; JX, Although only one contribution is listed, this analysis made an essential contribution justifying authorship. She approved the final version, Analysis and interpretation of data; AJ, ST, MKa, Acquisition of data, Contributed unpublished essential data or reagents; JZL, Conception and design, Analysis and interpretation of data; ZY, Acquisition of data, Analysis and interpretation of data, Drafting or revising the article, Contributed unpublished essential data or reagents; GJ, Conception and design, Acquisition of data, Analysis and interpretation of data, Drafting or revising the article, Contributed unpublished essential data or reagents; DJK, Conception and design, Analysis and interpretation of data, Drafting or revising the article

## Author ORCIDs

Jun Z Li, http://orcid.org/0000-0001-6727-0812
Gabor Juhasz, http://orcid.org/0000-0001-8548-8874
Jun Hee Lee, http://orcid.org/0000-0002-2200-6011
Daniel J Klionsky, http://orcid.org/0000-0002-7828-8118
Margit Burmeister, http://orcid.org/0000-0002-1914-2434

## Ethics

Human subjects: Study protocols including written informed consents have been approved by the University of Michigan Institutional Review Board and the Boğaziçi University Institutional Review Board for Research with Human Participants.

## Additional files

### Major datasets

The following datasets were generated:

| Author(s) | Year | Dataset title | Dataset URL | Database, license, and accessibility information |
|---|---|---|---|---|
| Sandford and Burmeister M | 2014 | Ataxia Gene Identification by Integrated Genomic Analysis | http://www.ncbi.nlm.nih.gov/projects/gap/cgi-bin/study.cgi?study_id=phs000757.v1.p1 | Publicly available at the NCBI (Accession no: phs000757.v1.p1). |

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
