## [Decision Letter]

Thank you for submitting your work entitled "Mutation in *ATG5* Reduces Autophagy and Leads to ataxia with Developmental Delay" for consideration in *eLife*. Your article has been favorably evaluated by a Senior Editor and three reviewers, one of whom, Noboru Mizushima, is a member of our Board of Reviewing Editors, and another is Joan Steffan.

The reviewers have discussed the reviews with one another and the reviewing editor has drafted this decision to help you prepare a revised submission.

Summary:

The authors identify a novel missense mutation in the *ATG5* gene in two ataxia patients. A partial reduction of the autophagic activity is observed in patients' cells as well as in yeast and fly having corresponding mutations. Locomotor abnormalities are also reproduced in fly mutants. This is the first report of a human disease having a pathogenic mutation in non-redundant core *ATG* genes.

Essential revisions:

Overall the data support the authors' conclusion. This is an important and timely study. However, as the authors performed exome sequencing of only one pair of patients, they should be careful to conclude that this is indeed a causative mutation.

1) To convince that the mutation in ATG5 is indeed "damaging", it is important to clearly show that the autophagic activity is reduced in mutant cells. However, the reduction is relatively small and even not convincing in some experiments. For example, in the bottom panel of Figure 3, bafilomycin A1-induced increase in the SQSTM1 level seems to be larger in ATG5 mutant cells compared to WT cells, suggesting that autophagic flux is not affected. On the other hand, the LC3-II turnover assay in Figure 3 clearly shows that the autophagic flux is reduced in ATG5 mutant cells. In Figure 3, rapamycin decreases the level of SQSTM1 in sample H but not in sample I (both are from unaffected individuals), making the evaluation of the patients' samples difficult. These inconsistencies need to be clarified. It would be recommended to include SQSTM1 blots and rapamycin (Torin 1 may be better) treatment in panel C. Including "rescued ATG5^E122D^ cells" expressing wild-type ATG5 would also strengthen the conclusion.

2) Along the same line, the autophagic activity is significantly reduced in mutant yeast cells at 3 h after starvation in Figure 5AB, but it is normal at 2-4 h and apparently reduced only at 6 h in Figure 5. This should be clarified. The authors use only the Pho8Δ60 assay to evaluate autophagy in yeast. It is recommended to use multiple methods to measure the autophagic activity (e.g., GFP-ATG8 cleavage, ATG8-PE formation, etc.).

---

## [Author Response]

Essential revisions: Overall the data support the authors' conclusion. This is an important and timely study. However, as the authors performed exome sequencing of only one pair of patients, they should be careful to conclude that this is indeed a causative mutation. 1) To convince that the mutation in ATG5 is indeed "damaging", it is important to clearly show that the autophagic activity is reduced in mutant cells. However, the reduction is relatively small and even not convincing in some experiments. For example, in the bottom panel of Figure 3, bafilomycin A1-induced increase in the SQSTM1 level seems to be larger in ATG5 mutant cells compared to WT cells, suggesting that autophagic flux is not affected. On the other hand, the LC3-II turnover assay in Figure 3 clearly shows that the autophagic flux is reduced in ATG5 mutant cells. In Figure 3, rapamycin decreases the level of SQSTM1 in sample H but not in sample I (both are from unaffected individuals), making the evaluation of the patients' samples difficult. These inconsistencies need to be clarified. It would be recommended to include SQSTM1 blots and rapamycin (Torin 1 may be better) treatment in panel C. Including "rescued ATG5^E122D^ cells" expressing wild-type ATG5 would also strengthen the conclusion.

We agree with the reviewers that the data presented in the former Figure 3 were confusing because of inconsistencies in the magnitude of autophagy defects. This is mainly due to the fact that the E122D mutation does not completely nullify the functionality of ATG5; therefore, a more careful experimental design was necessary.

In the former Figure 3, we subjected the cells to overnight treatments of rapamycin, thapsigargin and bafilomycin A_1_, which may induce secondary non-specific effects. For example, in addition to the autophagic mechanism, the level of SQSTM1/p62 is also controlled by a transcriptional mechanism (J Biol Chem. 2010 285:22576-91), which may be non-specifically altered by prolonged treatments with these drugs. Therefore, the experiment represented by the former Figure 3 was re-evaluated as a poor design, which would make the evaluation of patients’ samples difficult. Correspondingly, we have removed this figure panel and have instead performed the reviewers’ suggested experiment.

Figure 3 in the former manuscript (now Figure 3 in the revised manuscript) is a better designed experiment, and clearly shows that the autophagic flux is reduced in ATG5 mutant cells. As the reviewers recommended, we have included SQSTM1 blots and Torin 1 treatment in this set of data. The new results presented in Figure 3 of the revised manuscript convincingly demonstrate that (1) ATG12–ATG5 conjugates are reduced in E122D homozygotic patient cells, (2) SQSTM1/p62 is upregulated in these patient cells and (3) bafilomycin A_1_-induced LC3-II accumulation, which indicates autophagic flux, is attenuated in the patients’ cells, regardless of Torin 1 treatment. We think that these new experiments clarify that autophagic flux is indeed attenuated in ATG5^E122D^ mutant patient cells.

We also agree that rescuing patient cells with wild-type ATG5 would lend additional support concerning the disease-causing role of the E122D mutation. However, this experiment is challenging because of the extremely low transduction efficiency for LCL cells. These cells are not amenable to conventional transfection methods (PEI or liposomes). Even lentiviruses, which can transduce ~99% of monolayer cells such as HeLa, NIH3T3 and HepG2, were only able to transduce less than 5% of LCL suspensions (based on GFP expression). Though direct rescue of patient cells was impossible due to these technical difficulties, we have shown that human ATG5^WT^, but not ATG5^E122D^, is effective in restoring autophagy defects and neurodegenerative phenotypes exhibited in Atg5-knockout flies (Figure 6). We have also demonstrated that ATG5^E122D^ is biochemically defective in ATG12 conjugation in both cell culture and structural biology experiments (Figure 4). Therefore, the data strongly indicate that ATG5^E122D^ is functionally inferior to ATG5^WT^ in mediating autophagy, results that are also supported by the studies in yeast (see below).

*2) Along the same line, the autophagic activity is significantly reduced in mutant yeast cells at 3 h after starvation in Figure 5AB, but it is normal at 2-4 h and apparently reduced only at 6 h in Figure 5. This should be clarified. The authors use only the Pho8Δ60 assay to evaluate autophagy in yeast. It is recommended to use multiple methods to measure the autophagic activity (e.g., GFP-ATG8 cleavage, ATG8-PE formation, etc*.).

Again, the subtle variations between experimental results was the result of the hypomorphic nature of the E122D (E141D in yeast) mutation. By repeating the experiment (Figure 5 in the revised manuscript) using multiple time points, we have shown that the difference in autophagic activity, measured by the Pho8Δ60 assay, is statistically significant in later time points (4-6 h). Although not statistically significant, there is also a clear trend that the E141D mutation leads to a decreased Pho8Δ60 activity at earlier time points (1-3 h) as well. Considering that the Pho8Δ60 activity reflects a cumulative result of autophagic flux, it is actually not surprising that a longer incubation period would lead to a stronger statistical significance. Importantly, the same results are seen when we use strains with an integrated Atg5^E141D^ mutation (Figure 5).

Given the hypomorphic nature of the mutation, we agree that it is better to use multiple methods to measure the autophagic activity. Therefore, we performed GFP-ATG8 cleavage and ATG8–PE formation assays, as the reviewers recommended. In the GFP-ATG8 cleavage assay (J Biol Chem. 2004 279:29889-94), we have shown that the generation of free GFP, which is indicative of autophagic activity, is slightly but significantly attenuated in a yeast strain whose Atg5 was substituted with Atg5^E141D^ (Figure 5 in the revised manuscript).

We have also performed the Atg8‒PE formation assays. As shown by the ratio of Atg8–PE:total Atg8, cells expressing Atg5^E141D^ displayed a decrease in Atg8–PE conjugation compared to cells expressing Atg5^WT^ (Figure 5 in the revised manuscript). We also extended this analysis by using the in vivo reconstitution of Atg8–PE conjugation as described previously (J Cell Biol. 2008 182:703-13). In brief, we used a multiple-knockout (MKO) strain in which 23 *ATG* genes are deleted to examine Atg8 lipidation when expressing only the E1, E2 and E3-like conjugation enzymes. Atg8∆R that lacks the C-terminal arginine was used in the assay to bypass the initial activation step mediated by Atg4; due to the absence of Atg4, there is no cleavage of Atg8–PE from the membrane, resulting in stabilization of this form of the protein. We found that the Atg5^E141D^ mutant was strongly defective in Atg8–PE conjugation compared to the cells with Atg5^WT^ when Atg16 was not present (Figure 5 in the revised manuscript). As explained in the revised manuscript, Atg16 is not required mechanistically for Atg8 conjugation, but its presence increases the efficiency of this process and may dictate the site of conjugation (J Cell Biol. 2008 182:703-13). Thus, the presence of Atg16 (as in Figure 5) may partially mask the Atg8 lipidation defects of the Atg5^E141D^ mutant, and this would be at least one reason as to why the E122D/E141D mutation induces a hypomorphic rather than a complete null phenotype in both humans and model organisms.

Collectively, these new experiments performed in both yeast and mammalian cells, together with formerly included results from *Drosophila*, strongly support our conclusion that the E122D mutation is indeed damaging ATG5 function in mediating autophagy.